# Synthesis and Study of Zinc Orotate and Its Synergistic Effect with Commercial Stabilizers for Stabilizing Poly(Vinyl Chloride)

**DOI:** 10.3390/polym11020194

**Published:** 2019-01-23

**Authors:** Feng Ye, Qiufeng Ye, Haihua Zhan, Yeqian Ge, Xiaotao Ma, Yingying Xu, Xu Wang

**Affiliations:** 1College of Textile and Garment, Shaoxing University, Shaoxing 312000, China; zhh@usx.edu.cn (H.Z.); geyeqian@163.com (Y.G.); mxt775382687@163.com (X.M.); 18868914618@163.com (Y.X.); 2Ningbo Institute of Materials Technology and Engineering, Chinese Academy of Sciences, Zhejiang 315000, China; yeqiufeng@nimte.ac.cn; 3College of Chemical Engineering and Materials, Zhejiang University of Technology, Hangzhou 310014, China; wangxu@zjut.edu.cn

**Keywords:** poly(vinyl chloride), zinc orotate, dibenzoylmethane, thermal stabilizer, pentaerythritol, synergistic effect

## Abstract

Zinc orotate (ZnOr_2_), which is a new kind of poly(vinyl chloride) (PVC) stabilizer, is prepared in this work through the precipitation method, and its impact on the thermal stability of PVC is measured by thermogravimetric analysis (TG), Congo red test, and discoloration test. The results exhibit that the thermal stability of PVC is positively enhanced after the addition of ZnOr_2_. In contrast with a commercial thermal stabilizer, zinc stearate (ZnSt_2_), a noteworthy improvement was observed that ZnOr_2_ could postpone the “zinc burning” of PVC. This is principally ascribed to the Or anion in the structure of ZnOr_2_ being able to absorb the HCl released by PVC, and to supersede unstable chlorine atoms in the structure of PVC. In addition, blending ZnOr_2_ with calcium stearate (CaSt_2_) in diverse mass ratios can significantly accelerate the thermal stability of PVC. Optimum performance was achieved with a CaSt_2_:ZnOr_2_ ratio of 1.8:1.2. Moreover, an outstanding synergistic effect can be observed when CaSt_2_/ZnOr_2_ is coupled with other commercial auxiliary stabilizers. The initial color and long-term stability of PVC including CaSt_2_/ZnOr_2_ is significantly increased when pentaerythritol (PER) is added, while dibenzoylmethane (DBM) can only improve its long-term thermal stability.

## 1. Introduction

Poly(vinyl chloride) (PVC) is a kind of thermoplastic polymer manufactured by the polymerization of vinyl chloride monomer. It has extensive applications in the living and industrial fields because of its advantages for mature production technology, easy to process molding, corrosion resistance, abrasion resistance, flame retardant properties, and good electrical insulating performance [1,2,3]. However, the HCl can be easily removed from the structure of PVC, because the molecule contains allylic and tertiary chlorine atoms. The HCl will then further catalyze the degradation of PVC, resulting in the product gradually converting from white to completely black [4,5,6]. Therefore, thermal stabilizers should be added to PVC to prevent its degradation.

Over the past few decades, various compounds have been studied as PVC stabilizers, such as metal soaps [7], lead salts [8], and organotin [9,10,11]. Although lead salts and organotin have been shown to be highly effective in stabilizing PVC, the properties of high toxicity for lead salts and high price for organotin limit their application [9,10,11]. On the other hand, metal soaps are also restricted because of their low workpiece ratio. Therefore, it is urgently necessary to discover environmentally friendly and efficient thermal stabilizers. Recently, calcium/zinc thermal stabilizer (Ca/Zn), as one of the most commonly used metal soap stabilizers, has been playing an important role in this field due to its low toxicity [12,13]. However, an undesired product formed from Ca/Zn in PVC is zinc chloride (ZnCl_2_), which may accelerate the degradation of PVC and lead to a sudden “zipper dehydrochlorination” attributed to its strong Lewis acidity, so it can produce the “zinc burning” phenomenon [14]. Therefore, researchers have spent a considerable amount of time seeking a new zinc stabilizer that has no active catalytic effect on degrading PVC. Huanzhang Chen and co-workers synthesized zinc cyanurate under weak acid conditions, and investigated its thermal stability when combined with PVC. It was found that zinc cyanurate was superior to traditional zinc stearate [15]. Xu Wang and co-workers investigated the synergistic effect between zinc cyanurate and calcium stearate on the thermal stability of PVC [16]. Baoqing Shentu and co-workers researched the synthesis of zinc mannitol and its potency to stabilize PVC [17]. It was found that zinc mannitol could significantly enhance the thermal stability of PVC. Shumin Li and Youwei Yao studied the synthesis of zinc barbiturate, which showed good efficiency and synergistic effects on the thermal stability of PVC [18].

On the other hand, Starnes and co-workers carried out a large number of experiments to investigate pyrimidine diketone derivatives as PVC thermal stabilizers [19]. These compounds exhibited thermal stability for PVC, and some have been applied for industrial products. Simultaneously, Santamaria and co-workers also reported pyrimidine compounds as thermal stabilizers for PVC, and investigated the mechanism of thermal stability process [20]. It was found that pyrimidinedione derivatives could replace labile chlorines by *N*-alkylation reaction and stop the growth of polyene sequences in PVC chains. Therefore, we deduce that the presence of pyrimidine diketone structures in the compounds had an active effect in stabilizing PVC. In this article, we consider the structural resemblance of orotic acid (Or) to pyrimidinedione. The target of this work is to research the possibility of using zinc orotate (Zn(C_5_H_3_N_2_O_4_)_2_, abbreviated as ZnOr_2_) as a new thermal stabilizer for PVC.

## 2. Materials and Methods

### 2.1. Materials

PVC (SG-5, polymerization temperature: 55 °C, average degree of polymerization: 1000) applied in this work was acquired from Xinjiang Tianye (Group) Co. Ltd., Shihezi, China. Calcium stearate (CaSt_2_, calcium content: 6.6–7.4%), zinc stearate (ZnSt_2_, zinc content: 10–12%), and calcium carbonate (CaCO_3_, 1000 mesd) were acquired from Zhejiang Himpton New Material Co. Ltd., Hangzhou, China. Ca/Zn thermal stabilizers (Ca/Zn) were included in the form of CaSt_2_ (50 wt %) and ZnSt_2_ (50 wt %). Dioctyl phthalate (DOP, 98%), orotic acid (Or) (AR, 98%), zinc acetate (AR, 99%), dibenzoylmethane (DBM, AR), and pentaerythritol (PER, AR) were acquired from Aladdin Reagent., Shanghai, China. Other chemical reagents used in this study were analytical reagents. 

### 2.2. Preparation and Characterization of ZnOr_2_

ZnOr_2_ was prepared on the basis of the following process: Or (1.74 g, 10 mmol) and sodium hydroxide (0.4 g, 10 mmol) were mixed in a 100 mL three-neck flask with a magnetic drive stirrer, a condenser, and a dropping funnel. Next, 20 mL distilled water was added to clarify while stirring at room temperature. After dissolving zinc acetate (2.20 g, 10 mmol) with 15 mL of distilled water, the zinc acetate solution was slowly dripped into the Or solution. After all had been added, the mixed solution was reacted for 1 h at 85 °C. The filtrate was removed by extraction filtration and subsequently washed three times with distilled water. Finally, the white solid (ZnOr_2_) was dried in the oven at 180 °C, and the yield was 78.6%. The synthesis pathway of ZnOr_2_ is shown in Scheme 1.

The ZnOr_2_ was measured using an elemental analyzer (Euro EA 3000, EA Instruments, Milan, Italy) to determine the content of carbon, hydrogen, and nitrogen in ZnOr_2_. The zinc content of ZnOr_2_ was ascertained using the crucible process. The molecular structure of the sample was ascertained through Fourier transform infrared (IRPrestige-21, Shimadzu Corp., Kyoto, Japan) in the range of 4000–400 cm^−1^ with a KBr disc. The thermogravimetric curve was confirmed through thermogravimetric analyzer (TG/DTA6300, Seiko Instruments Inc., Chiba, Japan) in an air atmosphere, with temperature increasing from 25 to 800 °C at a heating rate of 10 °C/min.

### 2.3. Preparation of PVC Samples

Mixtures containing PVC powder resin (50 phr), PVC paste resin (50 phr), DOP (50 phr), CaCO_3_ (15 phr), and stabilizers (3 phr) were mixed in a beaker. After being stirred to homogeneity, the mixtures were poured into a glass mold with a thickness of 1.0 mm to plasticize for 40 min at the temperature of 140 °C. After the plasticization was completed, the mixture was cooled slightly and cut with an engraving knife to the proper size (15.0 mm × 20.0 mm × 1.0 mm).

### 2.4. Evaluation of Stabilizing Efficiency

#### 2.4.1. Congo Red Test

The PVC powder was ground and mixed with 3 phr thermal stabilizers in a mortar. Next, the Congo red test paper was placed in a glass tube containing the sample. The distance between the Congo red test paper and the PVC sample was 20 mm. The static thermal stability of the PVC sample was evaluated by immersing the glass tube in an oil bath at 180 °C. The static thermal stability time (T_s_) is defined as the time when the Congo red paper begins to turn blue. Congo red tests were carried out three times to accurately calculate the mean value of thermal stability time.

#### 2.4.2. Discoloration Test

PVC slices with approximately 1.0 mm thickness were cut to size, about 15.0 mm × 20.0 mm. They were then placed on an aluminum foil paper and shifted to a temperature-controlled ageing oven (UF260, Memmert Inc., Schwabach, Germany). The PVC sheets were heated at 180 °C. PVC sheets were removed from the ageing oven every 10 min and scanned with a scanner (LiDE120, Canon Inc., Tokyo, Japan). The effect of the thermal stabilizer on the PVC can be compared through the color change of the heated PVC sheets.

#### 2.4.3. Thermogravimetric Analysis

Thermal degradation of the stabilized PVC samples was tested on a thermogravimetric analyzer (TG/DTA6300, Seiko Instruments Inc., Chiba, Japan) from room temperature to 800 °C at a heating rate of 10 °C /min, in an air atmosphere.

#### 2.4.4. Investigation of the Mechanism of ZnOr_2_ for Stabilizing PVC

The subjacent two experiments were carried out to investigate the pattern of behavior of ZnOr_2_ as a thermal stabilizer for stabilizing PVC [16,21]. 

A certain amount of ZnOr_2_ was added to a three-necked flask and then heated to 180 °C in the oil bath. After being subjected to a steady flow of dry HCl gas for 2 h, the sample was subsequently heated to 120 °C in air for 4 h to get rid of the remaining HCl gas. The treated product was put into the deionized water and then filtered to remove filter residue. Finally, one drop of 0.1 mol/L silver nitrate solution was added to ascertain whether the filtrate included chloride ions. This experiment could ultimately demonstrate whether ZnOr_2_ can be taken as an HCl absorber.

Another experiment was conducted to explore whether ZnOr_2_ could supersede unstable chlorine atoms in PVC chains. A certain amount of ZnOr_2_ and PVC powder were ground and mixed in a mortar, and subsequently mixed in an open twin-wheel mill for 10 min at 180 °C. After mixing was completed, the PVC containing ZnOr_2_ was dissolved in tetrahydrofuran and filtered to get rid of the unreacted ZnOr_2_. Finally, the PVC samples were precipitated with methanol and gathered by filtration. The purified samples were aged at different time intervals (0–40 min) at 180 °C in air, and characterized by FTIR spectroscopy (IRPrestige-21, Shimadzu Corp., Kyoto, Japan).

## 3. Results

### 3.1. Characterization of ZnOr_2_

Or is an organic acid, so it is understood that the ZnOr_2_ synthesized in neutral conditions in our lab is an organic salt. The elemental analysis results of ZnOr_2_ revealed that the atomic concentration proportion of zinc to nitrogen was about 1:4. And the zinc content of ZnOr_2_ acquired by heat weight of the crucible was 15.68 %. Therefore, we infer that the molecular formula of this compound is Zn(C_5_H_3_N_2_O_4_)_2_·2H_2_O (theoretical value of zinc content is 15.82%) [22,23]. 

In this article, ZnOr_2_ is used to represent Zn(C_5_H_3_N_2_O_4_)_2_·2H_2_O. The thermal action of ZnOr_2_ was characterized by TGA at heating rate of 10 °C/min. As shown in Figure 1, the TGA curve of ZnOr_2_ reveals an estimated mass loss of 8.9% at the temperature range from 25 to 300 °C, which was assigned to the absorbed water. This result also proved that there are two absorbed water molecules in ZnOr_2_ [24,25].

It can also be observed from the curve that further disintegration of ZnOr_2_ occurred over 300 °C, which can be broken down into two stages. The first thermal weight loss stage occurred with an approximate mass loss of 64.9% (calculated 63.1%) at a temperature extent of 300–420 °C, which is mainly attributed to the release of three carbon monoxide molecules. The second mass loss stage of 8.42%, which occurred at the temperature extent of 490–800 °C, can be attributed to the decomposition of remnant cyanurate anions, and finally left the ZnO residue at 18.4% (calculated 19.71%). On the other hand, the TGA curve also suggests that ZnOr_2_ has very little weight loss when the temperature is dropped to 200 °C. Therefore, ZnOr_2_ is stable within the processing temperature of PVC systems, in the range of 160–200 °C.

The Fourier transform infrared (FTIR) spectra of ZnOr_2_ and Or are shown in Figure 2. As shown in Figure 2, It can be observed that the stretching vibration peak and symmetric stretching vibration peak of COOH for Or were transferred from 3515.4 and 1271.6 cm^−1^ to 3415.8 and 1373.9 cm^−1^, respectively, after generating Zinc salts. It can also be ascertained that ZnOr_2_ has been successfully synthesized [26,27].

### 3.2. Thermal Stability of ZnOr_2_ Stabilized PVC

Figure 3 shows the Congo red test results of PVC samples containing various thermal stabilizers [26,27]. From the figure, it can be observed that the thermal stability time of pure PVC is about 390 s. After addition of the ZnOr_2_ and Ca/Zn thermal stabilizers, the thermal stability times of PVC samples were increased to 500 s and 490 s, respectively. The results indicate that PVC samples stabilized with ZnOr_2_ have better thermal stability by comparison than those containing commercial Ca/Zn stabilizers. However, the thermal stability time of PVC samples descended to 340 s after adding 3 phr Or. This result indicates that the Or structure containing carboxyl groups was not able to absorb the HCl liberated by PVC, and the existence of free acidic groups will promote the degradation of PVC. The thermal stability of the zinc salt stabilizer used in this paper, ZnOr_2_, has a similar effect to the Ca/Zn commercial thermal stabilizer. Hence, it is clear that ZnOr_2_ has a strong ability to absorb HCl, and can be applied as an potential thermal stabilizer [27,28].

Figure 4 shows the effects of discoloration for PVC with different kinds of thermal stabilizers at 180 °C in air [27,28]. The PVC strip containing only 3 phr ZnSt_2_ showed outstanding initial color, but turned absolutely black within 10 min. The reason is that ZnSt_2_ can restrain discoloration by superseding the labile chlorine atoms on PVC chains and producing the undesirable ZnCl_2_. Moreover, the PVC sample will suddenly turn black with the increase of ZnCl_2_ content; this adverse effect on the long-term stability of PVC is called the “zinc burning” phenomenon. Furthermore, the complete discoloration time was delayed to 60 min for PVC samples stabilized with Ca/Zn (1:1) thermal stabilizer, which is mainly put down to the interaction between ZnCl_2_ and CaSt_2_. A remarkable improvement was observed in that ZnOr_2_ could postpone the complete discoloration time, contrasted with ZnSt_2_ and Ca/Zn (1:1) [29]. Meanwhile, as shown in Figure 3 and Figure 4, a PVC strip stabilized with ZnOr_2_ showed better initial colors than pure PVC, and the PVC strip stabilized with ZnSt_2_. This demonstrates that ZnOr_2_, just like ZnSt_2_, can also supersede the labile chlorine atoms and prevent the formation of conjugated double bonds in PVC chains in the preliminary stage of degradation. The formation of these bonds can result in the discoloration of PVC [30]. 

The TGA curves of the pure PVC and PVC stabilized with 3% stabilizers are shown in Figure 5. From the figure, we can see that the thermal degradation process of PVC can be divided into two stages. The first stage involves the evolution of hydrogen chloride, which occurs principally in the range of 180–380 °C. The second stage involves chain breaking and crosslinking, and the release of hydrogen chloride, aromatic compounds, and polyenic compounds, which occurs primarily in the range of 380–550 °C. The effect of thermal stabilizers is chiefly reflected at the first stage [31,32]. Table 1 summarizes the relevant data about initial decomposition temperature (T_5%_ and T_10%_), most rapid decomposition temperature (T_r_), and weight loss at the first stage (W_f_) from the TGA curves. It can be observed that the pure PVC retained 29.6% of its weight at the first stage (W_f_), and PVC stabilized by ZnOr_2_ or CaSt_2_/ZnOr_2_ retained more weight (about 35.7% and 35.9%, respectively) than pure PVC. The greater residual weight was put down to ZnOr_2_ absorbing the HCl liberated by the degradation of PVC [31,32]. Moreover, this phenomenon can also prevent the shortcoming of rapid degradation of PVC, because ZnOr_2_ can effectively absorb HCl, reduce its concentration, and then delay the rapid degradation of PVC. This result can explain why PVC with ZnOr_2_ has higher thermal stability.

For the sake of suggesting a conceivable mechanism which could clarify the stabilizing efficiency of ZnOr_2_, two experiments were executed to research the mode of behavior of ZnOr_2_ as a thermal stabilizer for PVC [33]. According to the first experiment, white sediment was observed in the pellucid filtrate when one drop of 0.1 mol/L silver nitrate solution was added in. This indicated the existence of chloride ions in the filtrate, and ultimately demonstrated that ZnOr_2_ could carry out a reaction with HCl at 180 °C. Moreover, Figure 6 gives the FTIR spectrum of ZnOr_2_ before and after treatment with HCl at 180 °C. The bands of the HCl-treated sample at 1523.8, 1329.5, and 983.9 cm^−1^ were put down to the asymmetric deformation vibrations, symmetric deformation vibrations, and rocking vibrations of NH^2+^, respectively. It can also be seen that the FTIR spectrum of the treated sample shows a band at 732.1 cm^−1^ specific for C–Cl, which confirms that ZnOr_2_ plays the part of an HCl absorber [34].

Another experiment was executed to research whether ZnOr_2_ could supersede the labile chlorine atoms on a PVC chain [30,34]. A certain amount of ZnOr_2_ and PVC powder was mingled on a two-wheel open mill for 10 min at 180 °C. After dissolution of the PVC stabilized with ZnOr_2_ into tetrahydrofuran and subsequent filtering separation, the unreacted ZnOr_2_ was eliminated. Finally, the PVC sample was precipitated through methanol and gathered by filtration. 

The FTIR spectra of ZnOr_2_, the purified PVC/ZnOr_2_ sample after ageing for 0 min, and pure PVC are shown in Figure 7a,c,d, respectively. It can be seen that the FTIR spectrum of the purified PVC/ZnOr_2_ sample after ageing for 0 min indicated the existence of a new peak at 1677.7 cm^−1^ corresponding to the (
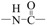
) group in ZnOr_2_. This feature indicates that chemical bonds form between the orotate anion in ZnOr_2_ and the degraded polymeric chains during the stabilization process. Moreover, the FTIR spectrum of the purified PVC/ZnOr_2_ sample after ageing for 40 min is shown in Figure 7b. It can be seen that this fresh peak becomes weak and cannot be observed after being aged for 40 min [28]. These results suggest that the (
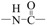
) group in ZnOr_2_ may play a major role in suppressing the thermal degradation of PVC.

According to the above-mentioned experimental results, a conceivable mechanism for PVC stabilized by ZnOr_2_ is suggested in Scheme 2. On the basis of the mechanism diagram, the unstable chlorine atom (allyl chloride) on the PVC chain is first separated as a chloride ion and then left behind a carbocation on the polymer chain (**1**). The separated chlorine ion is absorbed by ZnOr_2_, leading to formation of orotate anion intermediates and ZnCl_2_ (**2** and **3**). The orotate anion intermediates are appended to the positive charge generated on the PVC chain (**4**). This suggests that ZnOr_2_ can postpone further zipper degradation by superseding the labile chlorine atoms in PVC chains. In addition, the orotate anions in ZnOr_2_ have intense ability to absorb the hydrogen chloride liberated by the degradation of PVC (**5** and **6**), which will reduce the formation of ZnCl_2_. It has been declared that “zinc burning” occurs only after ZnCl_2_ concentrations reach a certain level. As a result, ZnOr_2_ provides higher long-term stabilizing efficiency than ZnSt_2_ and Ca/Zn thermal stabilizers.

### 3.3. Influence of Mixed Thermal Stabilizers on Thermal Stability of PVC

From the aforementioned study result, it can be observed that ZnOr_2_ has a better ability to replace unstable chlorine atoms and assimilate HCl than some commercial eco-friendly thermal stabilizers (ZnSt_2_, etc.), and so it can availably improve the thermal stability of PVC by itself. However, ZnOr_2_ cannot provide superior thermal stability compared with traditional lead salt stabilizers. Hence, in order to create more high-efficiency and eco-friendly thermal stabilizers, it should be integrated with other stabilizers (metal soap, auxiliary stabilizers, etc.) to manufacture complex thermal stabilizers [3,35]. This study includes an investigation of choosing calcium stearate (CaSt_2_) as a traditional metal soap stabilizer, with DBM and PER as auxiliary thermal stabilizer.

The results of the Congo red test for PVC with various concentration ratios of CaSt_2_/ZnOr_2_ and CaSt_2_/ZnSt_2_ thermal stabilizers are shown in Figure 8 and Figure 9, severally. It can be observed from Figure 8 that the thermal stability time of PVC/ZnOr_2_ was correspondingly short, but could be increased by increasing the CaSt_2_ concentration from 0.0 to 2.4. CaSt_2_ can react with ZnCl_2_ to produce CaCl_2_, which can suppress the catalytic degradation of PVC, so that the thermal stability time of the PVC samples was step by step improved with increasing CaSt_2_ content [3,35]. Interestingly, it was observed that the thermal stability time of PVC was decreased if only used with CaSt_2_. The reason is that there are good synergy effect between ZnOr_2_ and CaSt_2_, which can efficaciously enhance the thermal stability of PVC. For another, Figure 9 also shows similar testing results: that the thermal stability time of PVC/ZnSt_2_/CaSt_2_ was increased with the increasing of CaSt_2_ concentration [3,35]. By comparing Figure 8 and Figure 9, it can be confirmed that ZnOr_2_ displays a better stabilizing effect for PVC than the commercial thermal stabilizer ZnSt_2_.

The results of the oven discoloration test for PVC with various concentration proportions of CaSt_2_/ZnOr_2_ and CaSt_2_/ZnSt_2_ thermal stabilizers are shown in Figure 10 and Figure 11, severally [36]. From Figure 10, it can be observed that the PVC samples with various proportions of CaSt_2_/ZnOr_2_ thermal stabilizers showed consistent results with the Congo red test. The discoloration degree of PVC samples was also continuously delayed when the CaSt_2_ content was increased gradually. This result also can be considered to mean CaSt_2_ has an active effect for improving the long-term stability of PVC/ZnOr_2_ [28,37]. Entertainingly, it can be observed that the effect of thermal stability was highest when the content proportion of the CaSt_2_/ZnOr_2_ was 1.8:1.2. Compared with other proportions, it was found that the complete black time of PVC reached 190 min, as well as its initial whiteness being prominently improved. It was also proven that a good synergy effect took place between ZnOr_2_ and CaSt_2_ and availably enhanced the thermal stability of PVC. By comparing Figure 10 and Figure 11, it can be shown that the CaSt_2_/ZnOr_2_ complex stabilizer has a generally more stable effect than CaSt_2_/ZnSt_2_, which can ultimately indicate that ZnOr_2_ has a better stabilizing effect than the ZnSt_2_ thermal stabilizer.

It is well known that an auxiliary stabilizer should be added to Ca/Zn stabilizers to improve their thermal stability in practical application. Therefore, it was expected to produce a synergistic stabilization effect on PVC when CaSt_2_/ZnOr_2_ stabilizer was combined with the auxiliary stabilizers dibenzoylmethane (DBM) and pentaerythritol (PER), commonly used β-diketone and polyols, respectively. They are also the most frequently used organic auxiliary thermal stabilizers for PVC. To this end, these two commercial compounds were studied as auxiliary stabilizers for CaSt_2_/ZnOr_2_ stabilizers.

Figure 12 and Figure 13 respectively show PVC Congo red and oven discoloration test results with various ratios of CaSt_2_/ZnOr_2_/DBM thermal stabilizers. As shown in Figure 12, it can be observed that the stability time of PVC/CaSt_2_/ZnOr_2_/DBM samples was longer than PVC/CaSt_2_/ZnOr_2_ samples. Hence, it was indicated that addition of DBM could availably contribute to enhancing the stability time of PVC [28,37]. Moreover, it can be seen from Figure 13 that the initial whiteness of PVC samples was decreased, as well as the long-term stability being greatly enhanced, when auxiliary stabilizer DBM was added. It is suggested that there are several groups on the structure of DBM which have the capacity to supersede unstable chlorine atom as well as ZnOr_2_. A competition to replace unstable chlorine atoms on PVC occurred between DBM and ZnOr_2_, which led to the decrease of the initial whiteness of PVC samples. Furthermore, the long-term stability of the PVC was greatly improved after the addition of DBM. Metal complexes formed between DBM and ZnCl_2_, so that the ZnOr_2_ had more effectively capacity to absorb HCl. The synergy effect between DBM and ZnOr_2_ effectively improves the long-term stability of PVC. From the above mentioned, it can be confirmed that ZnOr_2_ provided valid initial stability as well as inferior long-term stability. Moreover, the addition of DBM to PVC/ZnOr_2_ is not necessary because DBM also plays an important role in inhibiting coloring ability of the PVC at early stages.

Figure 14 and Figure 15 respectively show PVC Congo red and oven discoloration test results with various ratios of CaSt_2_/ZnOr_2_/PER thermal stabilizers. As shown in Figure 14, it was observed that the stability times of PVC/CaSt_2_/ZnOr_2_/PER samples were higher than PVC/CaSt_2_/ZnOr_2_ samples. Hence, it is indicated that addition of PER can availably contribute to enhancing the stability time of PVC [28,37]. Moreover, the result was different with addition of DBM. It can be seen from Figure 15 that the initial whiteness of PVC samples maintained a relatively high level, while the long-term stability was markedly improved when auxiliary stabilizer PER was added. ZnOr_2_ possesses the ability to both replace unstable chlorine atoms and absorb HCl, while PER has the capacity to complex with ZnCl_2_. The synergistic effect between them effectively improved the initial stability and long-term stability of the PVC. Thus, it can be considered that PER is an appropriate auxiliary stabilizing agent for PVC/CaSt_2_/ZnOr_2_.

In order to ultimately prove the experimental results of the Congo red and oven discoloration tests, the relationship between weight and time of PVC, PVC/ZnOr_2_, PVC/CaSt_2_/ZnOr_2_, PVC/CaSt_2_/ZnOr_2_ /PER, and PVC/ZnSt_2_ was characterized by thermogravimetric analysis at a specific temperature about 220 °C for 2 h, and the results are summarized in Figure 16 [27,35]. It can be observed from the figure that the sequence for loss of thermal weight is as follows: PVC/CaSt_2_/ZnOr_2_/PER > PVC/CaSt_2_/ZnOr_2_ > PVC/ZnOr_2_ > PVC/ZnSt_2_. It can therefore be confirmed that the thermal stability of ZnOr_2_ is better than ZnSt_2_. Moreover, the thermal stability was greatly increased after the addition of CaSt_2_ and auxiliary stabilizer PER. The results of thermogravimetric analysis agreed with the Congo red and oven discoloration method. Therefore, it can be verified that both CaSt_2_ and PER take a synergistic effect with ZnOr_2_ to improve the thermal stability of PVC. Hence, the complex of CaSt_2_/ZnOr_2_/PER can be considered a high-efficiency thermal stabilizer.

## 4. Conclusions

ZnOr_2_ was successfully synthesized by a straightforward precipitation method, and characterized through elemental analysis, FTIR spectroscopy, and TGA. ZnOr_2_ was demonstrated to be an valid thermal stabilizer for PVC. Contrasted with ZnSt_2_, ZnOr_2_ can effectively postpone the “zinc burning” of PVC. After combining with CaSt_2_, the CaSt_2_/ZnOr_2_ complex can also improve the long-term stability of PVC remarkably. In order to improve the thermal stability of PVC, DBM and PER were added as auxiliary stabilizers into CaSt_2_/ZnOr_2_, to prepare complex thermal stabilizers. Both of the two auxiliary stabilizers displayed good long-term stability for PVC when combined with CaSt_2_/ZnOr_2_. Moreover, PER has a valid synergistic effect with CaSt_2_/ZnOr_2_ that enhances the initial whiteness of PVC. Therefore, CaSt_2_/ZnOr_2_/PER can be applied as a potential high efficiency complex thermal stabilizer for PVC in the future.

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
