# Peer review of "Synthesis and Study of Zinc Orotate and Its Synergistic Effect with Commercial Stabilizers for Stabilizing Poly(Vinyl Chloride)"

_polymers, 2019, doi:10.3390/polym11020194_

Round 1
Reviewer 1 Report
I am herewith sending the comments about polymers-431666.

Author Response
Dear Editors and Reviewers:
Thank you very much for your letter and for the reviewers’ comments concerning our manuscript entitled “Synthesis and study of zinc orotate and its synergistic effect with commercial stabilizers for poly (vinyl chloride)” (ID: polymers-431666). From the comments, I am clearly observed that my dear editors and reviewers are all very responsible and respected. Those comments are all valuable and very helpful for revising and improving our paper, as well as the important guiding significance to our researches. We have studied comments carefully and have made correction which we hope meet with approval. Revised portion are marked in red in the paper.

Reviewer 2 Report
I recommend to add to Introduction one more citation (concerning application of zinc cyanurate for PVC stabilization): J. Hebel Univ. Sci. Technol., 37(1), 33-38 (2016).
It is very well known that PVC is thermally stable only up to 160-165 oC and it should be applied at temparatures below 160 oC. While, experimental methods described in this manuscript were conducted at 180 oC. So, I have many doubts if Congo red test and discooration test are appropriate for this kind of studies. The PVC blends with stabilizers were thermally unstable at 180 oC in the above tests within longer time than 600-1000s and longer than 10 min., respectively.
Thermal stability of PVC-ZnOr2 blends was only slightly better, than obtained wit Zn and Ca stearates or their mixtures. Moreover, zinc orotate (ZnOr2), applied as a new PVC stabilizer is much more expensive than other traditional PVC stabilizers. Thus, it seems that the practical future meaning of ZnOr2 as a PVC stabilizer will be rather negligible.
Colors in Fig. 16, used for PVC-ZnOr2 and PVC-ZnSt2 blends, look the same, and should be different.
Author Response

(The authors gave the same response as above.)
